# The Associations of Neutrophil–Lymphocyte, Platelet–Lymphocyte, Monocyte–Lymphocyte Ratios and Immune-Inflammation Index with Negative Symptoms in Patients with Schizophrenia

**DOI:** 10.3390/biom13020297

**Published:** 2023-02-04

**Authors:** Marina Šagud, Zoran Madžarac, Gordana Nedic Erjavec, Ivona Šimunović Filipčić, Filip Luka Mikulić, Dunja Rogić, Zoran Bradaš, Maja Bajs Janović, Nela Pivac

**Affiliations:** 1School of Medicine, University of Zagreb, 10000 Zagreb, Croatia; 2Department of Psychiatry and Psychological Medicine, University Hospital Centre Zagreb, 10000 Zagreb, Croatia; 3Rudjer Boskovic Institute, 10000 Zagreb, Croatia; 4Faculty of Dental Medicine and Health, Josip Juraj Strossmayer University of Osijek, 31000 Osijek, Croatia; 5Department for Laboratory Diagnostics, University Hospital Centre Zagreb, 10000 Zagreb, Croatia

**Keywords:** negative symptoms of schizophrenia, NLR: nutrophil–lymphocyte ratio, PLR: platelet–lymphocyte ratio, MLR: monocyte–lymphocyte ratio, SII index: systemic immune-inflammation index, SCZ: schizophrenia, physical anhedonia, social anhedonia

## Abstract

Neutrophil–lymphocyte ratio (NLR), platelet–lymphocyte ratio (PLR), monocyte–lymphocyte ratio (MLR) and systemic immune-inflammation index (SII index) are increasingly used as indicators of inflammation in different conditions, including schizophrenia. However, their relationship with negative symptoms, including anhedonia, is largely unknown. Included were 200 patients with schizophrenia and 134 healthy controls (HC), assessed for physical anhedonia (PA), using the Revised Physical Anhedonia Scale (RPAS), and social anhedonia (SA) by the Revised Social Anhedonia Scale (RSAS). Patients were rated by the Positive and Negative Syndrome Scale (PANSS), the Clinical Assessment Interview for Negative Symptoms (CAINS) and the Brief Negative Symptom Scale (BNSS). Most of the negative symptoms were in a weak to moderate positive correlations with blood cell inflammatory ratios, namely, between NLR and MLR with PANSS negative scale, CAINS, and BNSS, and in male patients, between PLR and PANSS negative scale and CAINS. Fewer correlations were detected in females, but also in a positive direction. An exception was SA, given the negative correlation between its severity and the SII index in females, and its presence and higher PLR in males. While different negative symptoms were associated with subclinical inflammation, the relationship between SA and lower inflammatory markers deserves further exploration.

## 1. Introduction

Complete blood count cell parameters are widely used non-specific markers of systemic inflammation. While neutrophils are the first-line of defense during infection [1], increased neutrophil counts also indicate non-specific inflammation in many conditions, such as atherosclerosis, autoimmune disease and cancer [2]. Lymphocyte count reflects the specific immunity, generated to prevent new infections and respond to tumor cells, representing host immune competence [3]. Therefore, the neutrophil to lymphocyte ratio (NLR), calculated by dividing neutrophil count by lymphocyte count, provides a balance of two opposite, but complementary, pathways [4]. A higher NLR was associated with increased mortality in the general population [5] and in psychiatric patients [6].

Monocytes play a key role in the development of atherosclerotic plaques. They operate either directly, via antigen presentation and cytokine secretion, or through their differentiation into macrophages or foam cells which secrete pro-inflammatory cytokines and chemokines [7]. The combination of elevated monocyte and low lymphocyte counts into a monocyte to lymphocyte ratio (MLR) is a novel inflammatory parameter.

Platelets have an important role in hemostasis. Activated platelets release multiple substances, which influence many processes beyond blood clotting, such as inflammation, immunity, and tumor growth [8]. Higher platelet to lymphocyte ratio (PLR) is considered to reflect inflammation, atherosclerosis and platelet activation [9].

More recently, a novel, low-cost and accessible indicator of inflammation and immunity, the systemic immune-inflammation index (SII index), has been introduced [10]. It has been recognized as a superior marker that reflects the balance between host inflammatory and immune response status better than PLR, NLR and MLR [11]. It is calculated as (neutrophil × platelet)/lymphocyte ratio. An elevated SII index was associated with, among many other conditions, an increased risk of cardiovascular disorders [12].

Immunological changes in patients with schizophrenia (SCZ) are well established [13], including alterations in blood count cell numbers and ratios. Patients with SCZ [14,15] or non-affective psychosis [16] had a higher NLR, while a lower NLR was protective against SCZ [17]. In addition, patients with SCZ [18] and non-affective psychosis [16] had a higher MLR than healthy controls, while NLR, PLR and MLR were increased in first-episode patients [12]. Interestingly, in a machine-learning study, MLR and NLR were among the most discriminative features between patients with SCZ and healthy controls [19].

SCZ is a highly complex disorder. According to International Classification of Disease (ICD-11), it has positive, negative, psychomotor, cognitive and affective symptom dimensions. Negative symptoms are considered as a core component of SCZ [20]. They are associated with elevated inflammatory markers, mostly C-reactive protein (CRP), or different cytokines across the disease spectrum [21]. NLR did [22,23] or did not [17] correlate with the disease severity, and it was associated with positive, but not negative symptoms [22], or did not correlate with either dimension [23]. The discrepancies may arise from different disease stages since NLR may increase in acute phases, and decline in remission [24,25]. Those studies assessed negative symptoms evaluated by the Positive and Negative Syndrome Scale (PANSS) [17,22,23], which does not incorporate the latest research on negative symptoms [26]. Besides, anhedonia, as one of the key negative symptoms, is not included in PANSS items. Preclinical evidence suggests that early inflammation associated with SCZ might produce anhedonia [27], at least in male offspring [28], whereas higher granulocyte to lymphocyte ratio correlated with anhedonia in rats [29].

So far, there are no published studies on NLR and negative symptoms measured by scales other than PANSS, as well as on MLR or PLR and negative symptoms assessed by any clinical scale. In addition, there are no data on the relationship between NLR, MLR and PLR and anhedonia in SCZ. Despite being extensively investigated in numerous somatic diseases, only two studies examined SII in SCZ. The first reported no differences in SII between the first-episode psychosis and healthy controls [30], while the second found higher SII in SCZ than in healthy individuals [31]. In addition, due to a lack of such studies, the association between SII and any symptoms of SCZ is still not clear.

Given the importance of both negative symptoms and low-grade inflammation in SCZ, the aim of this study was to investigate the association between blood cell counts and their ratios, and the severity of distinct domains of negative symptoms and social and physical anhedonia in patients with SCZ. Based on the current literature, we hypothesized that the severity of negative symptoms, including the presence and severity of physical and social anhedonia, would correlate with the blood cell inflammatory ratios.

## 2. Methods

### 2.1. Participants

We used data from our previous trials [32,33] to conduct the separate analysis on the association of NLR, MLR, PLR and SII and negative symptoms, as well as physical and social anhedonia in patients with SCZ, and to compare the data with values in healthy controls. Thus, several of the methods presented here have been previously described [32,33]. Briefly, 302 patients were originally included from the Department for Psychiatry and Psychological Medicine at the University Hospital Centre Zagreb. However, blood cell count data at the time of assessment were available for 200 patients with SCZ. Therefore, the present study included patients with SCZ (*n* = 200). The key inclusion criteria were (1) the diagnosis of SCZ, (2) age between 18 and 65 years, and (3) participants who signed an informed consent document. Patients excluded had (1) current severe somatic disease, including those with extrapyramidal symptoms, such as Parkinson’s disease, (2) cognitive difficulties or severe psychotic, aggressive or psychomotor symptoms, that compromised the capacity to cooperate during the interview, (3) recent alcohol or psychoactive drug abuse or dependence (less than 3 months ago), (4) the known presence of any current or chronic infection, hematologic or autoimmune disease and/or (5) the presence of any other severe medical disorder, such as malignancies. 

Healthy participants (*n* = 134) were enrolled if they had (1) no current or recent clinically significant somatic disease, (2) no current of previous psychiatric disorder and (3) were currently taking no drugs. Healthy controls were recruited mostly by the hospital staff, who are well-known by the investigators.

### 2.2. Procedures

Diagnosis of SCZ was confirmed by a structured clinical interview [34], based on the DSM-IV-TR criteria. The severity of psychopathology was assessed by the PANSS [35]. This instrument is composed of positive (seven items), negative (seven items), and general psychopathology (16 items) subscales. Each item describes the symptom severity from 1 (absent) to 7 (most severe). Negative symptoms were further evaluated by the Clinical Assessment Interview for Negative Symptoms (CAINS) [36] and the Brief Negative Symptom Scale (BNSS) [37]. The CAINS is composed of nine items, which evaluate motivation and interest for (1) social, (2) vocational, and (3) recreational activities, and a four-item expression scale, which measures the deficits in expression. Each item is rated on a scale of 0 to 4, ranging from no impairment to severe deficit. The BNSS includes 13 items, which are categorized in five negative symptom subscales (anhedonia, asociality, avolition, alogia, and blunted affect), whereas the 6th subscale evaluates the lack of normal distress. All items are rated on a 0 to 7 scale, with the higher scores denoting greater deficiencies.

Physical anhedonia (PA) was assessed by the Revised Physical Anhedonia Scale (RPAS), and Social Anhedonia (SA) was measured by the Revised Social Anhedonia Scale (RSAS) [38]. Both instruments are self-rating scales, with all items presented in a “true” or “false” format. The RPAS evaluates hedonic deficits in food, sex, and other activities, and is composed of 61 items. The RSAS scale determines deficits in gaining pleasure from non-physical stimuli, such as involvement in social contact, and has 40 items. Depending on the answers, participants were considered to have PA if they had RPAS scores >20 in women and >28 in men, while the cut-off scores for SA were >16 for women and >20 for men. Rating of all scales was conducted by the same rater (ZM).

The severity of depressive symptoms was estimated by the Calgary Depression Scale for SCZ (CDSS) [39]. This scale is comprised of nine items, each graded from 0 (absent) to 3 (severe). A total CDSS score higher than six is indicative of the presence of a depressive episode. 

General functioning was scored by the Global Assessment of Functioning (GAF) [40]. This scale measures functioning in social, occupational and psychological areas, on a continuous scale from 1, representing the most severe dysfunction, to 100, referring to superior functioning.

### 2.3. Blood Collection

Three milliliters of blood were collected, as a part of routine procedures, by a simple venipuncture from the cubital vein, between 7.00 and 9.00 a.m., after an overnight fasting and tobacco abstinence for more than 12 h. Blood biochemical indicators (numbers of leukocytes, eosinophils, neutrophils, lymphocytes, monocytes, platelets) were detected with an automatic biochemical analyzer. The NLR, PLR, MLR and SII index were calculated from the numbers of the three cell types.

### 2.4. Statistical Analyses

Statistical evaluation was done using R Statistics 3.5.1. Since the Kolmogorov–Smirnov test indicated violation of a normal distribution, data were compared by non-parametric tests (Mann–Whitney U-test for two groups and Kruskal–Wallis H test with post-hoc Dunn’s test for three or more groups). Multiple linear regression was used to determine the effects of age, sex and diagnosis on blood cell counts, total antipsychotic dose and scores of psychological scales used for patients’ evaluation, while correlations between the studied parameters were analyzed with a Spearman’s rank test. Results were expressed as median and 25th (Q1) and 75th (Q3) percentiles. A priori determination of the sample size and post hoc computation of the achieved power was conducted by the G*Power 3.1 Software [41]. For the Kruskal–Wallis test with α = 0.05, power = 0.80 and medium effect size (0.25), the total desired sample size was 200. For the Mann–Whitney test with α = 0.05, power = 0.80 and medium effect size (0.50), the total desired sample size was 134. For a Spearman’s rank test with α = 0.05, power = 0.80 and medium effect size (0.30), the total desired sample size was 282 and for multiple linear regressions with α = 0.05, power = 0.80, medium effect size (0.15) and three predictors, the total desired sample size was 77. As the actual total sample size was 334, the power analysis confirmed the appropriate effect size.

## 3. Results

This study included 200 patients with SCZ (paranoid or residual type) and 134 healthy control individuals, whose clinical and demographic data are presented in Table 1 and Table 2, respectively. Subjects were matched for age and no difference (U = 10510.50; *p* = 0.081) in age was found between healthy control subjects and patients with SCZ. Additionally, no difference (χ^2^ = 2.37; *p* = 0.132) in the distribution of men (63% in SCZ and 54.4% in HC) and women (37% in SCZ and 45.6% in HC) between the two groups of subjects was found. On the other hand, smokers were significantly more frequent (χ^2^ = 13.73; *p* < 0.001) in patients with SCZ (53.5%) than in healthy control subjects (29.3%).

### 3.1. The Effects of Diagnosis, Age and Sex on Psychopathology

A multiple regression analysis was used to determine the effect of diagnosis, age and sex on the CAINS, BNSS, RPAS and RSAS total scores blood cell counts and total antipsychotic dose, presented as a chlorpromazine equivalent. Significant effect of sex on the severity of physical anhedonia was detected (F = 12.59; *p* < 0.001; R^2^ = 0.12; β_sex_ = −0.17; p_sex_ < 0.001). Further investigation revealed that among patients with SCZ more men than women did not have physical or social anhedonia (Table 1), while among healthy control subjects, men had higher RPAS scores (Table 2). Significant effect of age on the severity of social anhedonia was also detected (F = 10.43; *p* < 0.001; R^2^ = 0.10; β_age_ = 0.14; p_age_ = 0.020). More precisely, a weak positive correlation (σ = 0.019; *p* = 0.035) between age and RSAS scores in men with SCZ was found, indicating a slight progression of physical anhedonia with older age in this group of subjects. As expected, the diagnosis of SCZ had a significant effect on RPAS (F = 12.59; *p* < 0.001; R^2^ = 0.12; β_diagnosis_ = −0.25; p_diagnosis_ < 0.001) and RSAS (F = 12.59; *p* < 0.001; R^2^ = 0.12; β_diagnosis_ = −0.25; p_diagnosis_ < 0.001) scores, but also leukocyte (F = 9.28; *p* < 0.001; R^2^ = 0.08; β_diagnosis_ = −0.27; p_diagnosis_ < 0.001), lymphocyte (F = 1.77; *p* = 0.152; R^2^ = 0.02; β_diagnosis_ = −0.12; p_diagnosis_ = 0.035) and monocyte (F = 2.79; *p* = 0.041; R^2^ = 0.03; β_diagnosis_ = −0.02; p_diagnosis_ = 0.006) counts. Neither age nor sex affected the severity of CAINS, BNSS and PANSS scales or chlorpromazine equivalents, while the number of eosinophils and neutrophils was not affected by any variable (data available on request). However, sex was the only variable that significantly affected the platelet count (F = 5.53; *p* = 0.001; R^2^ = 0.05; β_sex_ = 0.21; p_sex_ < 0.001). In patients with SCZ, total antipsychotic dose, presented as a chlorpromazine equivalent, was in a weak positive correlation with CAINS total (σ = 0.15; *p* = 0.029), CAINSrec (σ = 0.17; *p* = 0.018), CAINSexp (σ = 0.16; *p* = 0.025), BNSS total (σ = 0.18; *p* = 0.012), BNSSavol (σ = 0.18; *p* = 0.010), BNSSblaf (σ = 0.15; *p* = 0.030), PANSS total (σ = 0.24; *p* = 0.001), PANSpos (σ = 0.21; *p* = 0.003), PANSSneg (σ = 0.20; *p* = 0.005) and PANSSgen (σ = 0.20; *p* = 0.004) scores. Total antipsychotic dose was not in significant correlation with RPAS, RSAS or blood cell counts (data available on request). 

Since the cut-off values for RPAS and RSAS differ for men and women, and based on the results of multiple regression analysis, all further analyses, including RPAS, RSAS and platelet count, were conducted for men and women separately.

### 3.2. Differences in Clinical and Laboratory Parameters between Males and Females

Demographic data and symptoms of SCZ were similar across the sexes. Males had a positive history of drug abuse more frequently (χ^2^ = 8.16; *p* = 0.004), while female patients had higher depression scores (U = 5459.00; *p* = 0.040), and twice as common PAs (χ^2^ = 6.61; *p* = 0.009) and SAs (χ^2^ = 8.07; *p* = 0.005) (Table 1).

In contrast to patients, in control subjects, both anhedonias were present in only few individuals, without differences between males and females (Table 2).

Comparison of blood cell counts between patients with SCZ and healthy control subjects revealed that patients with SCZ had higher leukocyte, eosinophil, neutrophil, lymphocyte and monocyte counts (Table 3). Platelet-related variables were analyzed separately by sex, and PLR was lower in males with SCZ than in healthy males, whereas no differences were detected in females (Table 3).

Males with SCZ had a higher RPAS (U = 1803.50; *p* = 0.004) and RSAS (U = 1873.50; *p* = 0.003) than control males. Likewise, females with SCZ had a higher RPAS (U = 1071.00; *p* < 0.001) and RSAS (U = 1143.00; *p* = 0.001) than females from the control group.

### 3.3. The Relationships between Blood Cell Inflammatory Ratios, and PA and SA

We further investigated whether inflammatory ratios could reflect the presence of SCZ with none or either type of anhedonia. Since there were only few healthy subjects whose RPAS or RSAS scores were bigger than cut-off values for physical or social anhedonia (Table 2); those subjects (*n* = 9) were not included in this analysis. Results showed that only PLR values were different between male healthy control subjects and patients with or without either type of anhedonia (Table 4). SCZ patients with social anhedonia had the lowest PLR value, significantly different when compared to healthy control subjects and SCZ patients with physical anhedonia. Additionally, SCZ patients without anhedonia had significantly lower PLR values than healthy men and SCZ patients with physical anhedonia.

In order to investigate the relationship between inflammatory ratios and pathophysiology indicators in patients with SCZ, we conducted the Spearman’s rank tests. The presence of correlations between cell count ratios and RPAS and RSAS scores in patients with SCZ and healthy control subjects are shown separately for men and women in Table 5. Results indicate a weak negative correlation between the severity of social anhedonia and SII index in women with SCZ, and a weak to moderate positive correlation between the severity of social anhedonia and SII index and PLR in HC.

### 3.4. The Relationships between Blood Cell Inflammatory Ratios and Other Negative Symptoms

Correlations between other psychological scales used for patients’ evaluation, as well as a chlorpromazine equivalent, and inflammatory ratios are shown in Table 6. Since platelet counts are different between men and women, correlation indicators, including PLR and SII index, are presented for men and women separately. As mentioned in Section 3.1, positive correlations between several clinical scales/subscales and the total antipsychotic dose were obtained. In order to avoid any bias caused by patients’ therapy, CAINS total, CAINS_rec_, CAINS_exp_, BNSS total, BNSS_avol_, BNSS_blaf_, PANSS total, PANSS_pos_, PANSS_neg_ and PANSS_gen_ scores were corrected for the effect of total antipsychotic dose by fitting the linear model of each score in dependence of total antipsychotic dose, and using the obtained residuals for the correlation analysis between psychological scales and blood cell inflammatory ratios. Most of the tested variables were in weak to moderate positive correlation, with the exception of chlorpromazine equivalent, which was found to be negatively correlated with PLR in women with SCZ (Table 6).

## 4. Discussion

The present results revealed associations between higher peripheral inflammatory ratios and many different negative symptoms of SCZ. A vast majority of associations were positive, suggesting the link between the severity of specific negative symptoms and inflammation in both sexes, although more prominent in males. The only exception was social anhedonia, since we have found inverse correlations between its presence and PLR in male, and between its severity and SII index in female patients with SCZ. 

Patients with SCZ had higher leukocyte, neutrophil, lymphocyte, eosinophil and monocyte counts than healthy individuals, but men with SCZ had a lower PLR than healthy men.

### 4.1. Anhedonia and Blood Cell Inflammatory Ratios

Despite comparable levels of psychopathology in males and females, as reflected by similar PANSS, CAINS, BNSS total scores and their subscales, as well as GAF scale scores, both anhedonias were twice as common in females. Such differences were not observed in healthy controls which, indeed, had a very low prevalence of both anhedonias. 

The higher prevalence of both anhedonias in female patients could not be attributed to a higher severity of anhedonia or greater negative symptoms, but rather to lower cut-off RPAS and RSAS scores for corresponding anhedonias in females [42]. Moreover, female patients had slightly higher CDSS scores, and previous research reported the correlation between both physical and social anhedonias with CDSS in SCZ [43]. In addition, anhedonia may be associated with other features in SCZ which have not been assessed in the present study, such as fatigue [44]. There is also evidence on the higher sensitivity to anhedonia, at least in healthy females, such as decreased ventral striatum activity during reward anticipation following endotoxin injection, which was not observed in males [45]), and lower sensitivity to reward and higher sensitivity to punishment than males [46]. Those findings emphasize sex differences in physical and social anhedonia across populations. This is also supported by preclinical data, which demonstrated that sex differences in anhedonia-/depressive-like behavior were accompanied by molecular leads [47].

PA was associated with higher PLR values in male patients, while the intensity of BNSS-measured anhedonia was positively associated with SII index in males, and NLR in females, suggesting higher markers of inflammation in PA and BNSS-measured anhedonia. Such findings are in line with recent evidence supporting the relationship between inflammation and anhedonia [48,49], and a preclinical study in a rat model of depression, which reported the correlation between NLR and lower sucrose preference scores [29]. In clinical settings, anhedonia may also affect healthy habits. For example, higher severity of anhedonia in patients with SCZ was associated with a poor adherence to a Mediterranean diet [50], and a low adherence to a Mediterranean diet was associated with higher PLR [51].

Unexpectedly, male patients with SA had a lower PRL than healthy controls, while female patients with higher social anhedonia severity had a lower SII index, implicating the connection between lower inflammatory indices and SA in SCZ. Such results emphasize distinct characteristics of SA among other negative symptoms, regarding its relationship with inflammatory blood cell parameters. Anhedonia is a complex [52], transdiagnostic [33] concept, consisting of several domains, which may not be equally impaired in SCZ. For example, recent meta-analysis reported that individuals with SCZ endorse greater challenges, experiencing social rewards relative to physical rewards [53].

Lower values of some markers of inflammation related to SA in patients with SCZ may be associated with the antipsychotic use. This may also help explain the different directions of correlations between females from SCZ and HC groups, given that healthy females had weak to moderate positive correlation between severity of SA, and SII index and PLR. Of note, antipsychotic medications reduce symptoms of SCZ, but may not reduce anhedonia, which is supposed to be a trait marker. For example, in a preclinical model, quetiapine treatment reduced inflammation but did not reduce anhedonia [54]. However, in contrast to our study, inflammation was determined by the cytokine levels [54]. Moreover, in our study, total antipsychotic dose was inversely correlated with PRL in females, suggesting the connection between antipsychotic dose and lower PLR. In agreement, a decrease in platelet count was detected following treatment of SCZ, and PLR decreased in the group of patients who responded to treatment [55]. Given that our patients were in a stable condition, and male patients had an even lower PRL than healthy males, we cannot exclude the contribution of treatment to such PRL values. Another explanation may include lower exposure to infections in people with SA, due to the avoidance of social contacts and consequent social isolation, which may have resulted in lower inflammatory indices. Namely, SA is associated with smaller and less diverse social networks, and fewer highly active social domains [56], while social network size is associated with a higher prevalence of upper respiratory tract infections [57]. However, other negative symptoms might also contribute to the social withdrawal: lack of motivation, reduced interaction, decreased energy, flat or blunted affect, and loss of interest [58]. Another reason may be associated with the low physical activity in relation to SA. For example, schizophrenic patients who had higher anhedonia levels did not participate when offered aerobic training, in contrast to those with low anhedonia levels [59]. In turn, exercise was reported to increase platelet count, and to decrease lymphocyte count, thus affecting PLR and NLR [60]. It must be mentioned, however, that in the former trial [59], anhedonia was assessed by the Self-assessment Anhedonia Scale which is not specific only for SA. However, aerobic exercise was also reported to reduce inflammatory markers in healthy adults [61], whereas negative symptoms are associated with sedentary behavior in patients with psychosis [62].

### 4.2. Other Negative Symptoms and Blood Cell Inflammatory Ratios

In the present study, the PANSS negative score was positively correlated with MLR and NLR, and only in males, also with PRL. Contrary to our findings, NLR was previously negatively associated with the severity of negative symptoms [63]. The discrepancies may result from the different assessment of negative symptoms and distinct illness phases. In the latter study, patients were assessed with the Brief Psychiatric Rating Scale [63], which measures only blunted effects, emotional withdrawal and motor retardation, and may not detect other negative symptoms, such as deficits in motivation and pleasure. In addition, this study included patients who were in the acute phases or in remission [63], while our patients, with their PANSS scores lower than 70, had moderately low psychopathology [64]. Other studies did not detect associations between NLR and the PANSS negative scores [17,22,23], but unlike the present study, reported correlations with PANSS positive symptoms [22]. These studies had much smaller samples [17,22,23]. In addition, the discrepancies may arise from different disease severities (i.e., Kovács et al. [22] included patients who had higher PANSS positive scores than our patients), and patients had a higher NLR than controls [17,23], which was not the case in our study. So far, no study reported the correlations between MLR, PLR and SII index and symptom severity in SCZ.

Nevertheless, our findings implicate the role of inflammation in negative symptoms. Given the association between the presence of one or more PANSS-measured negative symptoms with a higher prevalence of metabolic syndrome, and greater number of comorbidities [65] and increased cardiovascular disease risk factors in SCZ [66], our results suggest that chronic inflammation, particularly in relation to negative symptoms, may contribute to such adverse outcomes. Further prospective studies should determine whether blood cells ratio baseline values, and their relationships with negative symptoms, may predict cardiovascular morbidity later in life.

While a positive correlation was observed between NLR and MLR with the CAINS and BNSS total scores, and few motivational subscales, none of the blood cell ratios correlated with the CAINS expression subscale scores, suggesting the association of inflammation with motivational, but not expression domains. In fact, the relationship between motivational CAINS items and real-world functioning was previously observed in SCZ [67] and a negative association was demonstrated between motivational, but not expressive dimension, and functional outcome [68], and lower CAINS social factor scores were associated with a higher likelihood of social media use [69]. Those findings collectively suggest that low-grade inflammation may adversely affect motivation, which, in turn, might impair functionality in patients with SCZ.

Females had a lower number of correlations, but those few were also in the positive direction. PLR was correlated with the BNSS asociality subscale scores and with the CAINS recreational subscale scores. Higher asociality scores correlated with lower functionality in individuals with SCZ [70]. In the present study, asociality was the only BNSS item which correlated with all inflammatory ratios. Asociality was also the only BNSS item associated with abnormal functional connections in SCZ, given its negative correlation with fractional anisotropy in several brain regions, including the left orbital prefrontal and posterior cingulate cortex [71]. Elevated PLR was associated with the increased risk of thromboembolism [9] and patients with SCZ had a higher risk of deep-vein thrombosis and pulmonary thromboembolism [72]. Additional research is needed to establish the link between PLR, asociality, and adverse cardiovascular outcomes in this population. While sex differences in the link between blood cell ratios and negative symptoms were not thoroughly investigated in SCZ, there is a general agreement that estrogens improve inflammation, at least related to metabolic dysfunction [73], and both human and animal studies reported the association of menopause or ovariectomy with the rise of peripheral inflammatory markers [74]. Given the median age of 41 years in our females, and the average age of menopause of 48 in Croatian females [75], the impact of estrogens on the relationship between negative symptoms and inflammatory ratios cannot be excluded.

An increased NLR was associated with some biological findings in SCZ. In patients with early psychosis, NLR was associated with a leukocyte discoidin domain receptors (DDR1) hypermethylation, which was increased in patients compared to healthy controls [76]. In patients with SCZ, but not in healthy controls, NLR has positively correlated with the total oxidative status and oxidative stress index [23]. Peripheral inflammatory markers, such as inflammatory cytokines secreted from monocytes/macrophages, could access the brain, and contribute to negative symptoms by decreasing the activation of the ventral striatum and its connectivity with ventral medial prefrontal cortex [21]. In view of the cross-sectional design of the present trial, it is not clear if the inflammation affected brain regions associated with negative symptoms, or, on the other hand, negative symptoms, such as asociality, through the lack of activity, contributed to the inflammation. 

### 4.3. Limitations

Since this study employed a cross-sectional design, the causality cannot be explored. The scales measuring negative symptoms did not differentiate whether symptoms were primary or secondary [20]. Since the primary negative symptoms are the core disease symptoms, secondary negative symptoms may originate from the positive symptoms, antipsychotic treatment, social deprivation, and depressive symptoms [77]. However, our participants did not have prominent depression or positive symptoms, according to their low CDSS and PANSS scores. While social deprivation cannot be excluded, about a third of the patients were employed (data not shown) and none resided in the institution. The influence of antipsychotic medication on the development of secondary negative symptoms cannot be excluded, but more than 60% of the patients received only atypical antipsychotics, and the chlorpromazine equivalent dose was not high, which makes patients less likely to experience extrapyramidal side-effects, sedation, or other dose-related adverse events. Other limitations were a small sample size, and the lack of measurement of additional inflammatory markers, such as CRP or proinflammatory cytokines. However, the NLR was higher in participants with SCZ with elevated CRP, than in those with normal CRP levels [78], and correlated with CRP levels [79]. In fact, the NLR showed a better predictive value for inflammation comparable to that of the CRP [4].

## 5. Conclusions

Our findings highlight the use of more detailed negative symptom scales to detangle the complex association between negative symptoms and inflammatory markers. Given the detrimental effects of negative symptoms and their poor response to antipsychotics, more data are needed to better understand their role in inflammation, and to confirm if negative symptoms may be a potential target for anti-inflammatory treatment. In agreement, recent meta-analysis reported the beneficial effects of add-on treatment with anti-inflammatory medication to antipsychotics, including the improvement of negative symptoms [80]. Longitudinal studies are needed to establish the relationship between reliable peripheral inflammatory indices, such as NLR, MLR, PLR and SII index, [1] and treatment outcomes in SCZ. Future studies are also needed to establish if SA stands out as the only negative symptom which is inversely associated with inflammation.

## Figures and Tables

**Table 1 biomolecules-13-00297-t001:** Demographic and clinical data of patients with SCZ. Data are presented as a number (percentage) or median (Q1; Q3).

		Men	Women
Age (Years)		40 (33; 47)	41 (35; 50)
		U = 5231.00; *p* = 0.150	
Age when first diagnosed (years)		23 (18; 29)	25 (19; 29)
		U = 5103.50; *p* = 0.263	
Smoking	No	56 (44.4%)	37 (50.0%)
	Yes	70 (55.6%)	37 (50.0%)
		χ^2^ = 0.58; *p* = 0.447	
History of alcohol abuse	No	107 (84.9%)	68 (91.9%)
	Yes	19 (15.1%)	6 (8.1%)
		χ^2^ = 2.07; *p* = 0.150	
History of drug abuse	No	84 (66.7%)	63 (85.1%)
	Yes	42 (33.3%)	11 (14.9%)
		χ^2^ = 8.16; *p* = 0.004	
Antipsychotic dose (mg/day) *		600 (375; 980)	625 (375; 950)
		U = 4694.00; *p* = 0.935	
BNSS score		29 (22; 36)	29 (21; 39)
		U = 4601.00; *p* = 0.877	
BNSS_anh_		6 (4; 9)	7 (5; 10)
		U = 5120.50; *p* = 0.243	
BNSS_asoc_		4 (3; 5)	4 (2; 4)
		U = 4122.50; *p* = 0.158	
BNSS_avol_		6 (4; 8)	6 (3; 8)
		U = 4660.50; *p* = 0.997	
BNSS_alog_		2 (1; 4)	2 (0; 4)
		U = 4401.50; *p* = 0.502	
BNSS_blaf_		8 (6; 11)	9 (7; 12)
		U = 5122.00; *p* = 0.243	
CAINS score		23 (17; 27)	22 (16; 28)
		U = 4653.00; *p* = 0.982	
CAINS_soc_		6 (4; 8)	6 (4; 7)
		U = 4281.00; *p* = 0.332	
CAINS_voc_		2 (0; 6)	2 (0; 5)
		U = 4665.50; *p* = 0.993	
CAINS_rec_		4 (3; 6)	4 (3; 6)
		U = 4997.00; *p* = 0.389	
CAINS_exp_		7 (5; 9)	7 (5; 9)
		U = 4816.00; *p* = 0.695	
CDSS score		2 (0; 4)	3 (1; 7)
		U = 5459.00; *p* = 0.040	
GAF score		50 (41; 61)	45 (40; 61)
		U = 4073.00; *p* = 0.135	
PANSS score		69 (62; 79)	69 (61; 83)
		U = 4930.00; *p* = 0.498	
PANSS_pos_		12 (11; 16)	13 (11; 16)
		U = 4830.50; *p* = 0.669	
PANSS_neg_		21 (18; 23)	20 (17; 25)
		U = 4621.50; *p* = 0.918	
PANSS_gen_		35 (31; 40)	36 (31; 44)
		U = 5084.50; *p* = 0.284	
RPAS score		18 (13; 23)	16 (10; 21)
		U = 3909.00; *p* = 0.057	
Physical anhedonia	No	104 (82.5%)	49 (66.2%)
	Yes	22 **(17.5%)**	25 **(33.8%)**
		χ^2^ = 6.61; *p* = 0.009	
RSAS score		12 (8; 16)	13 (7; 17)
		U = 4787.00; *p* = 0.751	
Social anhedonia	No	108 (85.7%)	51 (68.9%)
	Yes	18 **(14.3%)**	23 **(31.1%)**
		χ^2^ = 8.07; *p* = 0.005	

* in chlorpromazine equivalents; BNSS-The Brief Negative Symptom Scale; CAINS-Clinical Assessment Interview for Negative Symptoms; CDSS-The Calgary Depression Scale for SCZ; GAF-Global Assessment of Functioning; PANSS-Positive and Negative Syndrome Scale; RPAS-Revised Physical Anhedonia Scale; RSAS-Revised Social Anhedonia Scale; significant results are indicated in bold.

**Table 2 biomolecules-13-00297-t002:** Demographic and clinical data of healthy individuals. Data are presented as a number (percentage) or median (Q1; Q3).

		Men	Women
Age		37 (26; 55)	34 (22; 51)
		U = 1512.50; *p* = 0.218	
Smoking	No	29 (70.7%)	34 (68.0%)
	Yes	12 (29.3%)	16 (32.0%)
		χ^2^ = 0.08; *p* = 0.0.779	
RPAS score		14 (10; 18)	11 (8; 14)
		U = 693.00; *p* = 0.016	
Physical anhedonia	No	38 (92.7%)	46 (95.8%)
	Yes	3 (7.3%)	2 (4.2%)
		χ^2^ = 0.41; *p* = 0.520	
RSAS score		9 (6; 12)	7 (5; 11)
		U = 902.00; *p* = 0.405	
Social anhedonia	No	40 (97.6%)	45 (91.8%)
	Yes	1 (2.4%)	4 (8.2%)
		χ^2^ = 1.39; *p* = 0.238	

RPAS-Revised Physical Anhedonia Scale; RSAS-Revised Social Anhedonia Scale.

**Table 3 biomolecules-13-00297-t003:** Differences in blood cell counts and their ratios between patients with SCZ and healthy control subjects. Data are expressed as median (Q1; Q3).

Blood Cell Type/Ratio		Blood Cell Count (×10^9^/L)	
		SCZ	Healthy Control
Leukocytes		7.3 (6.3; 9)	6.5 (5.8; 7.5)
		U = 9491.00; ***p* < 0.001**	
Eosinophils		0.2 (0.1; 0.3)	0.1 (0.1; 0.2)
		U = 11226.50; ***p* = 0.019**	
Neutrophils		4.1 (3.3; 5.7)	3.6 (3.1; 4.3)
		U = 10506.60; ***p* = 0.001**	
Lymphocytes		2.2 (1.8; 2.8)	2.0 (1.7; 2.3)
		U = 11251.00; ***p* = 0.010**	
Monocytes		0.6 (0.5; 0.8)	0.5 (0.5; 0.7)
		U = 9821.50; ***p* < 0.001**	
NLR		1.9 (1.4; 2.8)	1.8 (1.5; 2.3)
		U = 12477.50; *p* = 0.270	
MLR		0.3 (0.2; 0.4)	0.3 (0.2; 0.3)
		U = 12221.00; *p* = 0.173	
Platelets	Men	229 (198; 272)	229 (201; 259)
		U = 4210.00. *p* = 0.855	
	Women	262 (221; 297)	259 (232; 294)
		U = 2337.50; *p* = 0.722	
SII index	Men	430.2 (297.1; 684.1)	386.9 (323.6; 510.0)
		U = 3778.00; *p* = 0.218	
	Women	497.9 (332.5; 781.3)	518.6 (368.0; 650.9)
		U = 2297.00; *p* = 0.860	
PLR	Men	**99.4** (75.3; 129.0)	**114.0** (95.3; 129.4)
		U = 5195.00; ***p* = 0.024**	
	Women	127.2 (97.7; 158.1)	128.2 (109.4; 148.6)
		U = 2320.00; *p* = 0.781	

NLR-neutrophil to lymphocyte ratio (neutrophil count/lymphocyte count); MLR-monocyte to lymphocyte ratio (monocyte count/lymphocyte count); SII index-systemic immune-inflammation index (platelet count × neutrophil count/lymphocyte count); PLR-platelet lymphocyte ratio (platelet count/lymphocyte count). Significant results are indicated in bold.

**Table 4 biomolecules-13-00297-t004:** Inflammatory ratios compared between patients with SCZ, with or without anhedonia, and healthy control subjects. Data are expressed as median (Q1; Q3).

Diagnosis	Anhedonia	Inflammatory Ratio			
		SII Index	NLR	PLR	MLR
**Men**					
SCH	None	425.9 (294.9; 678.4)	1.8 (1.4; 2.8)	98.2 (75.3; 126.6)	0.3 (0.2; 0.4)
	Both	478.8 (429.2; 593.4)	2.0 (1.8; 2.7)	100.0 (70.9; 125.4)	0.3 (0.2; 0.4)
	Social	351.0 (273.4; 684.1)	1.6 (1.0; 2.4)	**85.2** (68.2; 100.0)	0.3 (0.3; 0.5)
	Physical	430.2 (388.4; 995.8)	2.7 (1.6; 3.6)	**121.4** (108.1; 147.5)	0.4 (0.2; 0.5)
HC without anhedonia		386.7 (323.3; 506.2)	1.7 (1.4; 2.2)	**112.6** (95.1; 129.4)	0.3 (0.2; 0.3)
		H = 4.86; *p* = 0.302	H = 5.88; *p* = 0.208	H = 11.78; ***p* = 0.019 ***	H = 3.65; *p* = 0.456
**Women**					
SCH	None	603.8 (360.5; 831.1)	2.0 (1.4; 2.9)	133.7 (100.0; 153.5)	0.2 (0.2; 0.3)
	Both	402.8 (278.3; 622.7)	1.7 (1.2; 2.5)	108.4 (83.5; 150.3)	0.2 (0.2; 0.3)
	Social	427.8 (354.3; 547.2)	1.7 (1.3; 2.4)	121.3 (104.5; 178.6)	0.3 (0.3; 0.4)
	Physical	501.5 (310.2; 685.2)	1.9 (1.6; 2.5)	131.5 (82.9; 158.1)	0.3 (0.2; 0.3)
HC without anhedonia		518.6 (368.0; 657.2)	2.0 (1.5; 2.4)	126.6 (109.4; 145.3)	0.3 (0.2; 0.3)
		H = 3.15; *p* = 0.533	H = 1.10; *p* = 0.894	H = 0.94; *p* = 0.919	H = 4.65; *p* = 0.325

SCH-SCZ; HC-healthy control; SII index-systemic immune-inflammation index (platelet count × neutrophil count/lymphocyte count); NLR-neutrophil to lymphocyte ratio (neutrophil count/lymphocyte count); PLR-platelet lymphocyte ratio (platelet count/lymphocyte count); MLR-monocyte to lymphocyte ratio (monocyte count/lymphocyte count). Significant results are indicated in bold. * post-hoc Dunn’s test *p* < 0.005 for: SCZ + social anhedonia vs. HC, SCZ + social anhedonia vs. SCZ + physical anhedonia, SCZ without anhedonia vs. HC, SCZ without anhedonia vs. SCZ + physical anhedonia. Significant results are indicated in bold.

**Table 5 biomolecules-13-00297-t005:** Correlation between the severity of social and physical anhedonia and inflammatory ratios in patients with SCZ.

	Correlation	SCZ				Healthy Control			
		PLR	SII Index	NLR	MLR	PLR	SII Index	NLR	MLR
		**Men**							
MRPAS	σ	0.10	0.10	0.11	0.16	−0.03	−0.05	−0.04	−0.08
	*p*	0.276	0.296	0.223	0.081	0.849	0.776	0.823	0.611
RSAS	σ	0.17	0.16	0.08	0.13	0.06	−0.02	−0.02	−0.04
	*p*	0.061	0.084	0.393	0.142	0.713	0.891	0.902	0.793
		**Women**							
RPAS	σ	0.03	−0.07	−0.01	−0.06	−0.15	−0.06	−0.12	−0.14
	*p*	0.831	0.579	0.918	0.594	0.314	0.711	0.408	0.346
RSAS	σ	−0.12	**−0.28**	−0.22	0.00	**0.30**	**0.36**	0.28	0.16
	*p*	0.328	**0.018**	0.065	0.995	**0.037**	**0.012**	0.051	0.271

SII index-systemic immune-inflammation index (platelet count × neutrophil count/lymphocyte count); NLR-neutrophil to lymphocyte ratio (neutrophil count/lymphocyte count); PLR-platelet lymphocyte ratio (platelet count/lymphocyte count); MLR-monocyte to lymphocyte ratio (monocyte count/lymphocyte count). RPAS-Revised Physical Anhedonia Scale; RSAS-Revised Social Anhedonia Scale. Significant correlations are indicated in bold.

**Table 6 biomolecules-13-00297-t006:** Correlation of inflammatory ratios with the severity of positive and negative symptoms and total antipsychotic dose presented as chlorpromazine equivalents in patients with SCZ.

	Correlation	Men		Women		Men and Women	
		PLR	SII Index	PLR	SII Index	NLR	MLR
CAINS	σ	**0.26**	**0.26**	0.22	0.14	**0.19**	**0.14**
	*p*	**0.004**	**0.004**	0.057	0.221	**0.008**	**0.047**
CAINS_soc_	σ	**0.22**	**0.25**	0.09	0.06	**0.15**	0.09
	*p*	**0.012**	**0.006**	0.427	0.621	**0.038**	0.203
CAINS_voc_	σ	0.03	0.01	−0.06	−0.08	0.00	−0.07
	*p*	0.745	0.949	0.601	0.520	0.965	0.309
CAINS_rec_	σ	0.16	**0.23**	**0.29**	**0.25**	**0.20**	0.12
	*p*	0.071	**0.010**	**0.012**	**0.029**	**0.005**	0.089
CAINS_exp_	σ	0.14	0.16	0.12	0.07	0.13	**0.15**
	*p*	0.131	0.082	0.314	0.570	0.067	**0.037**
BNSS	σ	0.17	**0.20**	0.22	0.12	**0.17**	**0.15**
	*p*	0.064	**0.027**	0.061	0.323	**0.018**	**0.036**
BNSS_anh_	σ	0.12	**0.20**	0.17	0.16	**0.18**	0.13
	*p*	0.188	**0.024**	0.157	0.178	**0.014**	0.071
BNSS_asoc_	σ	**0.26**	**0.22**	**0.31**	**0.25**	**0.20**	**0.16**
	*p*	**0.004**	**0.015**	**0.008**	**0.034**	**0.005**	**0.020**
BNSS_avol_	σ	0.09	0.04	0.07	−0.01	0.01	−0.01
	*p*	0.298	0.646	0.536	0.904	0.861	0.894
BNSS_alog_	σ	0.15	**0.20**	0.09	0.02	**0.15**	**0.20**
	*p*	0.098	**0.026**	0.471	0.849	**0.033**	**0.005**
BNSS_blaf_	σ	0.10	0.14	0.13	0.06	0.12	0.10
	*p*	0.249	0.126	0.290	0.621	0.098	0.155
PANSS	σ	0.06	−0.01	0.14	0.11	0.02	0.04
	*p*	0.474	0.875	0.228	0.361	0.754	0.596
PANSS_pos_	σ	−0.05	−0.10	0.09	0.08	−0.08	−0.04
	*p*	0.577	0.272	0.465	0.522	0.259	0.579
PANSS_neg_	σ	**0.21**	0.17	0.16	0.11	**0.15**	**0.14**
	*p*	**0.017**	0.060	0.185	0.339	**0.035**	**0.044**
PANSS_gen_	σ	0.03	−0.04	0.13	0.10	0.02	0.02
	*p*	0.740	0.680	0.281	0.402	0.826	0.831
Chlorpromazine equivalent	σ	−0.06	0.08	**−0.24**	−0.18	−0.01	0.08
	*p*	0.495	0.364	**0.040**	0.127	0.878	0.235

SII index-systemic immune-inflammation index (platelet count × neutrophil count/lymphocyte count); NLR-neutrophil to lymphocyte ratio (neutrophil count/lymphocyte count); PLR-platelet lymphocyte ratio (platelet count/lymphocyte count); MLR-monocyte to lymphocyte ratio (monocyte count/lymphocyte count); RPAS-Revised Physical Anhedonia Scale; RSAS-Revised Social Anhedonia Scale; CAINS-Clinical Assessment Interview for Negative Symptoms; CAINS_soc_-CAINS social subscale; CAINS_voc_-CAINS vocational subscale; CAINS_rec_-CAINS recreational subscale; CAINS_exp_-CAINS expretion subscale; BNSS-The Brief Negative Symptom Scale; BNSS_anh_-BNSS anhedonia subscale; BNSS_asoc_-BNSS asociality subscale; BNSS_avol_-BNSS avolition subscale; BNSS_alog_-BNSS alogia subscale; BNSS_blaf_-BNSS blunted affect subscale. PANSS-Positive and negative syndrome scale; PANSS_pos_-PANSS positive subscale; PANSS_neg_-PANSS negative subscale; PANSS_gen_-PANSS general psychopathology subscale. Significant correlations are indicated in bold.

## Data Availability

The data that support the findings of this study are available on request from the corresponding author (N.P.).

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
