# Peer review of "The Associations of Neutrophil–Lymphocyte, Platelet–Lymphocyte, Monocyte–Lymphocyte Ratios and Immune-Inflammation Index with Negative Symptoms in Patients with Schizophrenia"

_biomolecules, 2023, doi:10.3390/biom13020297_

Round 1

Reviewer 1 Report

This study reanalyse previous data with focus on "simple" blood parameters reflecting inflammation and their correlation with the symptoms. Although it is not emphasize in the introduction/abstract, but gender differences were also analysed in details. The topic is interesting and further knowledge. The manuscrip was properly written in general. However, the discussion was hard to follow and contained a lot of repetition of the results focusing mainly on correlation. I suggest to reorganize it along another domain (e.g. inflammation in SCZ, sex and SCZ)

Although the Tables are correct, but graphical presentation of the main results would help to get the message more easily. Therefore I recommend to add one figure with graphs of teh major differences, correlations.

Specific comments:

Spaces e.g. line 30 too much " PLR: Plate-"; line 71 missing "spectrum[21]." Line 133 "(RSAS)[38]"

Line 101: "the original included 302 patients" should be modified as "the originally included 302 patients were"

Line 104: "patients with schizophrenia (N=200)." Modify as "200 patients with schizophrenia."

Line 105-112 Please, be consistent in usage of comma ("schizophrenia, (2)", but "interview (3)")

Line 115 Were the healthy control somehow matched to patients? Please, add one sentence to result about comparing general parameters of the healthy and patient group (e.g. no significant difference was between them in general parameters).

Line 139 "All rating scales" perhaps "Rating of all scales"

Line 149 delete "after blood tests"

Line 157 "escape from" My suggestion "violated the"

Line 171 The posthoc power analysis confirmed not the sample size, but the effect size.

 Table 1 is almost identical to 10.3390/cimb43020045 Table 1.

Line 181 As far as I understood correctly, the analysis included diagnosis, thus, also healthy individuals. Thus, the sex age etc. effects are valid also for healthy individuals, while posthoc comparison described only patients (referring to Table 1 e.g. but not Table 2).

Line 183 Try to reformulate the sentence from negative ("did not have") to affirmative.

Line 187 "in this group of subjects" does it mean schizophrenic men? Or men in general?

Line 192 and what about diagnosis?

Line 199 It seems to me that based on 3.1. sex differences came up as an important difference. Therefore, a separate analysis was conducted and 3.2 appear, despite it was not formulated as a goal in the introduction. Please, clarify more precisely, why sex was analysed here separately.

4.1. Add reference and discuss sex difference. Try to shorten the repetition of the results.

Line 312 It was hard to pick up the meaning of this paragraph, what it wanted to discuss?

Line 331 "corelated"

Line 352 Add "previously"

Line 389 delete comma "inflammation, may"

Line 410-423 should be somehow merged with line 343-349 both dealing with the role of physical activity.

Line 450 Reformulate the sentence! In present form it seems the relationship between the blood markers should be studied.

I suggest using abbreviation for schizophrenia (e.g. SCZ) throughout the manuscript.

Author Response

Spaces e.g. line 30 too much " PLR: Plate-"; line 71 missing "spectrum[21]." Line 133 "(RSAS)[38]" Answer: thank you, we have done this.

Line 101: "the original included 302 patients" should be modified as "the originally included 302 patients were"

Answer: thank you, we have done this.

Line 104: "patients with schizophrenia (N=200)." Modify as "200 patients with schizophrenia."

Answer: thank you, we have done this.

Line 105-112 Please, be consistent in usage of comma ("schizophrenia, (2)", but "interview (3)")

Answer: thank you, we have done this.

Line 115 Were the healthy control somehow matched to patients? Please, add one sentence to result about comparing general parameters of the healthy and patient group (e.g. no significant difference was between them in general parameters).

Answer: We agree with this comment. The following added in the text (line 118): Healthy controls were recruited mostly from the hospital staff, which are well-known to investigators. Moreover, (line 182) in results, the following was inserted: 

Answer: We agree with this comment. Additionally, no difference (χ2=2.37; p=0.132) in the distribution of men (63 % in SCZ and 54.4 % in HC) and women (37 % in SCZ and 45.6 % in HC) between two groups of subjects was found. On the other hand, smokers were significantly more frequent (χ2=13.73; p<0.001) in patients with schizophrenia (53.5 %) than in healthy control subjects (29.3 %).

Line 139 "All rating scales" perhaps "Rating of all scales"

Answer: We agree with this comment. “All rating scales” was changed to “Rating of all scales”

Line 149 delete "after blood tests"

Answer: We agree with this comment and have deleted this.

Line 157 "escape from" My suggestion "violated the"

Answer: We agree with this comment: we have changed it as requested.

Line 171 The posthoc power analysis confirmed not the sample size, but the effect size.

Answer: Thank you for your comment. We have corrected as you suggested: ‘As the actual total sample size was 334, the power analysis confirmed the appropriate effect size.’

Table 1 is almost identical to 10.3390/cimb43020045 Table 1.

Answer: in the present study, the data are reported from the 10.3390/cimb43020045 (Madzarac et al, 2021) study, as mentioned in the “Methods”, but only a subset of 200 out of 302 patients were analyzed (those who have done the white blood cell tests). Therefore, table 1 in both manuscripts represents the same rating scales, but the average total scores are not identical.

Line 181 As far as I understood correctly, the analysis included diagnosis, thus, also healthy individuals. Thus, the sex age etc. effects are valid also for healthy individuals, while posthoc comparison described only patients (referring to Table 1 e.g. but not Table 2).

Answer: Thank you for your comment. We added a further explanation related to healthy individuals: ‘Further investigation revealed that among patients with schizophrenia more men than women did not have physical or social anhedonia (Table 1), while among healthy control subjects men had higher RPAS scores (Table2).’

Line 187 "in this group of subjects" does it mean schizophrenic men? Or men in general?

Answer: Thank you for your comment. It helped us to realize that we wrote RPAS instead of RSAS. The reported result refers to men with schizophrenia so we corrected the sentence as follows: ‘More precisely, a weak positive correlation (σ=0.019; p=0.035) between age and RSAS scores in men with schizophrenia was found, indicating a slight progression of physical anhedonia with older age in this group of subjects.’

Line 192 and what about diagnosis?

Answer: We agree with this comment. Thank you for your comment. Since CAINS, BNSS, PANSS and chlorpromazine equivalents are available only for schizophrenia patients, the effect of diagnosis was not tested for those variables. The number of eosinophils and neutrophils was not affected by the diagnosis, sex or age.

We have explained more precisely in the text (section 3.1.): ‘Neither age nor sex affected the severity of CAINS, BNSS and PANSS scales or chlorpromazine equivalents, while the number of eosinophils and neutrophils was not affected by any variable (data available on request).’’

Line 199 It seems to me that based on 3.1. sex differences came up as an important difference. Therefore, a separate analysis was conducted and 3.2 appear, despite it was not formulated as a goal in the introduction. Please, clarify more precisely, why sex was analysed here separately.

Answer: We agree with this comment. A multiple regression revealed significant effect of sex on the severity of physical anhedonia and the platelet count. Therefore, regarding the blood cell ratios, all variables which included platelets (platelets, PRL and SII index) were analyzed separately in men and women, while others were not.  Anhedonia was also analyzed by sex, given that both physical and social anhedonia were more common in female patients.

4.1. Add reference and discuss sex difference.

Answer: We agree with this comment. Thank you, the following was added in the “Discussion” (line 344):  The higher prevalence of both anhedonia in female patients could not be attributed to higher severity of anhedonia or greater negative symptoms, but rather to lower cut-off RPAS and RSAS scores for corresponding anhedonia in females (Kwapil et al, 2022). Moreover, female patients had slightly higher CDSS scores, and previous research reported the correlation between both physical and social anhedonia with CDSS in schizophrenia (Kollias et al, 2008). In addition, anhedonia may be associated with other features in schizophrenia which have not been assessed in the present study, such as fatigue (Laraki et al, 2023). There is also evidence on the higher sensitivity to anhedonia, at least in healthy females, such as decreased ventral striatum activity during reward anticipation following endotoxin injection, which was not observed in males (Moieni et al, 2019) and lower sensitivity to reward and higher sensitivity to punishment than males (Dhingra et al, 2021). Those findings emphasize sex differences in physical and social anhedonia across populations. This is also supported by preclinical data, which demonstrated that sex differences in anhedonia-/depressive-like behavior were accompanied by molecular leads (Paden et al, 2020).

The following literature was added to “References”:

Kollias CT, Kontaxakis VP, Havaki-Kontaxaki BJ, Stamouli S, Margariti M, Petridou E. Association of physical and social anhedonia with depression in the acute phase of schizophrenia. Psychopathology. 2008;41(6):365-70. DOI: 10.1159/000152378

Laraki Y, Bayard S, Decombe A, Capdevielle D and Raffard S (2023) Preliminary evidence that fatigue contributes to anhedonia in stable individuals diagnosed with schizophrenia. Front. Psychiatry. 14:1098932.  https://doi.org/10.3389/fpsyt.2023.1098932

Moieni M, Tan KM, Inagaki TK, Muscatell KA, Dutcher JM, Jevtic I, Breen EC, Irwin MR, Eisenberger NI. Sex Differences in the Relationship Between Inflammation and Reward Sensitivity: A Randomized Controlled Trial of Endotoxin. Biol Psychiatry Cogn Neurosci Neuroimaging. 2019;4(7):619-626. doi: 10.1016/j.bpsc.2019.03.010

Kwapil, T.R.; Crump, R.A.; Pickup, D.R. Assessment of Psychosis Proneness in African-American College Students. J. Clin.Psychol. 2002, 58, 1601–1614.  DOI: 10.1002/jclp.10078

Dhingra I, Zhang S, Zhornitsky S, Wang W, Le TM, Li CR. Sex differences in neural responses to reward and the influences of individual reward and punishment sensitivity. BMC Neurosci. 2021;22(1):12.  DOI: 10.1186/s12868-021-00618-3

Paden, W., Barko, K., Puralewski, R. et al. Sex differences in adult mood and in stress-induced transcriptional coherence across mesocorticolimbic circuitry. Transl Psychiatry 10, 59 (2020). https://doi.org/10.1038/s41398-020-0742-9

Try to shorten the repetition of the results.

Answer: We agree with this comment. Line 432: Nevertheless, the positive correlations between PANSS negative scores and NLR and MLR, as well as PLR and SII index in our male patients implicate the role of inflammation in negative symptoms”, was replaced by “Nevertheless, our findings implicate the role of inflammation in negative symptoms”.

Line 442: “In addition to positive correlation between NLR and MLR with the CAINS and BNSS total scores, and the PANSS negative symptoms, similar associations were further observed between NLR and 2 out of 3 CAINS motivational subscales, whereas MLR correlated with recreational CAINSS subscale scores. Likewise, in men PLR correlated with social, and SII index with social and recreational subscales, and in women PRL was associated only with recreational subscale. None of the blood cell ratios correlated with the CAINS expression subscale scores, was  replaced by: While positive correlation was observed between NLR and MLR with the CAINS and BNSS total scores, and few motivational subscales, none of the blood cell ratios correlated with the CAINS expression subscale scores,

Line 457: Females had lower number of PLR and SII index correlations, was replaced by: Females had lower number of correlations,..

Line 312 It was hard to pick up the meaning of this paragraph, what it wanted to discuss? This paragraph was re-formulated into the following text: Unexpectedly, male patients with SA had lower PRL than healthy controls, while female patients with higher social anhedonia severity had lower SII index., implicating the connection between lower inflammatory indices and SA in schizophrenia. Such results emphasize distinct characteristics of SA among other negative symptoms, regarding its relationship with inflammatory blood cell parameters. Anhedonia is a complex [46], transdiagnostic [33] concept, consisted of several domains, which may not be equally impaired in schizophrenia. For example, recent meta-analysis reported that individuals with schizophrenia endorse greater challenges experiencing social rewards relative to physical rewards [47]”.

Line 331 "corelated"

Answer: We agree with this comment and have changed that.

Line 352 Add "previously" Answer: We agree with this comment and have changed that.

Line 389 delete comma "inflammation, may" Answer: We agree with this comment and have deleted that.

Line 410-423 should be somehow merged with line 343-349 both dealing with the role of physical activity.The text from lines 410-423 was merged with text in lines 343-349 and changed in the following sentence:

Answer: We agree with this comment and have changed that. “However, aerobic exercise was also reported to reduce inflammatory markers in healthy adults [69], whereas negative symptoms are associated with sedentary behavior in patients with psychosis [70].”

Line 450 Reformulate the sentence! In present form it seems the relationship between the blood markers should be studied.

Answer: We agree with this comment and have changed that. Thank you, we re-formatted this sentence to the following: “Longitudinal studies are needed to establish the relationship between reliable peripheral inflammatory indices such as NLR, MLR, PLR and SII index, [1] and treatment outcomes of schizophrenia”.

I suggest using abbreviation for schizophrenia (e.g. SCZ) throughout the manuscript. Answer: We agree with this comment and have used this abbreviation.

Reviewer 2 Report

I want to commend you for your work on this important topic of association of immunoinflammation markers with negative symptoms in patients with schizophrenia. The following are some suggestions to improve the article:

1. Schizophrenia is a highly heterogeneous disease. What kind of schizophrenia nosological forms were predominantly included in the study?

2. The authors should more clearly define the hypotheses of the study.

3. What was the duration of antipsychotic treatment in the patients enrolled in the study? 

4. The authors should investigate the effect of drug treatment, or at least, correct for this bias, using appropriate methods (such as applying a general linear model). 

Author Response

Reviewer No  2

I want to commend you for your work on this important topic of association of immunoinflammation markers with negative symptoms in patients with schizophrenia. The following are some suggestions to improve the article:

  1. Schizophrenia is a highly heterogeneous disease. What kind of schizophrenia nosological forms were predominantly included in the study?

Answer: We agree with this comment and have written: “Patients had paranoid or residual schizophrenia. It was added in the text (line 179): “This study included 200 patients with SCZ (paranoid or residual type) and 134 healthy control individuals.”

  1. The authors should more clearly define the hypotheses of the study.

Answer: We agree with this comment and have added hypothesis (line 96): Based on the current literature, we hypothesized that the severity of negative symptoms, including the presence and severity of physical and social anhedonia, would correlate with the blood cell inflammatory ratios.

  1. What was the duration of antipsychotic treatment in the patients enrolled in the study?

Answer: Males were diagnosed with schizophrenia on average 17 years ago (given their current median age of 40, and mean age when first diagnosed of 23), and females were diagnosed with schizophrenia on average 16 years ago (given their current median age of 41, and mean age when first diagnosed of 25). We have no data if they received antipsychotics continuously since being diagnosed with schizophrenia, but if we assume so, we can say that the average duration of antipsychotic treatment was 17 years for males and 16 years for females.

  1. The authors should investigate the effect of drug treatment, or at least, correct for this bias, using appropriate methods (such as applying a general linear model).

Answer: We agree with this comment and have written, and thank you for your comment. We have accepted your suggestion and checked the relationship between all the studied parameters (CAINS, BNSS and PANSS scales and subscales, blood cell counts, RPAS and RSAS) and total antipsychotic dose, presented as chlorpromazine equivalent and found positive correlation of several psychological scales and subscales with total antipsychotic dose. In accordance with this finding, we have changed the manuscript as follows:

In section 3.1. we have added: ‘In patients with schizophrenia total antipsychotic dose, presented as chlorpromazine equivalent was in a weak positive correlation with CAINS total (σ=0.15; p=0.029), CAINSrec (σ=0.17; p=0.018), CAINSexp (σ=0.16; p=0.025), BNSS total (σ=0.18; p=0.012), BNSSavol (σ=0.18; p=0.010), BNSSblaf (σ=0.15; p=0.030), PANSS total (σ=0.24; p=0.001), PANSpos (σ=0.21; p=0.003), PANSSneg (σ=0.20; p=0.005) and PANSSgen (σ=0.20; p=0.004) scores. Total antipsychotic dose was not in significant correlation with RPAS, RSAS or blood cell counts (data available on request).’

In section 3.3. we added: ‘As mentioned in section 3.1., positive correlations between several clinical scales/subscales and the total antipsychotic dose were obtained. In order to avoid any bias caused by patients’ therapy, CAINS total, CAINSrec, CAINSexp, BNSS total, BNSSavol, BNSSblaf, PANSS total, PANSSpos, PANSSneg and PANSSgen scores were corrected for the effect of total antipsychotic dose by fitting the linear model of each score in dependence of total antipsychotic dose and using the obtained residuals for the correlation analysis between psychological scales and blood cell inflammatory ratios.’ Also, Table 6 was corrected accordingly.
